# Malnutrition Screening and Assessment

**DOI:** 10.3390/nu14122392

**Published:** 2022-06-09

**Authors:** Carlos Serón-Arbeloa, Lorenzo Labarta-Monzón, José Puzo-Foncillas, Tomas Mallor-Bonet, Alberto Lafita-López, Néstor Bueno-Vidales, Miguel Montoro-Huguet

**Affiliations:** 1Intensive Care Unit, Department of Medicina, University Hospital San Jorge, 22004 Huesca, Spain; jlabarta@unizar.es (L.L.-M.); tomas.mallor@gmail.com (T.M.-B.); lafita.alberto@gmail.com (A.L.-L.); nesbue@gmail.com (N.B.-V.); 2Faculty of Health and Sports Sciences, University of Zaragoza, 50009 Zaragoza, Spain; 3Clinical Analysis and Biochemistry Service, Department of Medicina, University Hospital San Jorge, 22004 Huesca, Spain; jpuzo@unizar.es; 4Unit of Gastroenterology, Hepatology, and Nutrition, Department of Medicina, University Hospital San Jorge, 22004 Huesca, Spain

**Keywords:** nutrition screening tools, malnutrition, nutritional assessment

## Abstract

Malnutrition is a serious problem with a negative impact on the quality of life and the evolution of patients, contributing to an increase in morbidity, length of hospital stay, mortality, and health spending. Early identification is fundamental to implement the necessary therapeutic actions, involving adequate nutritional support to prevent or reverse malnutrition. This review presents two complementary methods of fighting malnutrition: nutritional screening and nutritional assessment. Nutritional risk screening is conducted using simple, quick-to-perform tools, and is the first line of action in detecting at-risk patients. It should be implemented systematically and periodically on admission to hospital or residential care, as well as on an outpatient basis for patients with chronic conditions. Once patients with a nutritional risk are detected, they should undergo a more detailed nutritional assessment to identify and quantify the type and degree of malnutrition. This should include health history and clinical examination, dietary history, anthropometric measurements, evaluation of the degree of aggression determined by the disease, functional assessment, and, whenever possible, some method of measuring body composition.

## 1. Introduction

Nutrition is a basic life process that consists of taking in nutrients from our environment and using them to perform our vital functions including growth, reproduction, and the maintenance of our body, in sickness and in health. The nutritional stages are ingestion, digestion, absorption, transport, assimilation, and excretion of the waste products.

Malnutrition is a major health problem that can be caused by a primary situation, such as poverty, due to lack of food, or by a secondary situation, resulting from disease. Different mechanisms can be involved in secondary malnutrition: reduced intake because of the anorexia that accompanies the disease, and the metabolic stress caused by that, or as a consequence of the different treatments. This response to stress speeds up the metabolism, causing a hormonal imbalance that leads to an increase in protein catabolism, which consumes our protein reserves, altering the function of different organs and the activity of our immune defenses.

According to ESPEN, malnutrition, or undernutrition, is defined as “a state resulting from lack of intake or uptake of nutrition that leads to altered body composition (decreased fat free mass) and body cell mass leading to diminished physical and mental function and impaired clinical outcome from disease”. It can result from undernutrition, with or without catabolism, produced by the inflammatory state of both acute and chronic diseases. [1]. Paraphrasing Soeters: “Malnutrition is a subacute or chronic state of nutrition, in which a combination of varying degrees of undernutrition and inflammatory activity has led to changes in body composition and diminished function” [2].

Malnutrition is prevalent in many diseases, and especially in hospitalized patients, institutionalized elderly patients, and chronic patients [3]. The incidence of malnutrition in hospitalized patients is quantified at between 20% and 50%, depending on the diagnostic method used [4]. The consequences of malnutrition are a reduction in quality of life, as well as an increase in morbidity, the appearance of infections, poor wound healing, functional alterations in immune defense, a reduction in overall muscle strength, especially in pulmonary ventilation, and increased mortality, length of hospital stay, and hospital costs [5,6,7,8,9]. However, malnutrition is preventable if the problem is diagnosed early. Unfortunately, this is often not the case, due to poor awareness, information and knowledge, or a lack of protocols in place to identify it.

A systematic approach to addressing malnutrition in hospitals should begin with a nutritional risk assessment of all patients at admission, followed by a detailed assessment of the nutritional status of patients most at risk [10]. An appropriate nutritional intervention, tailored to the individual needs of patients identified as malnourished or at nutritional risk, should be implemented. Unfortunately, although the need for this process is fully acknowledged, it is not systematically implemented [11]; 21,000 patients from 325 hospitals in 25 European countries are included in a study by the “NutriDay” survey, with the results showing that only 52% (ranged between 21% and 73%) of the hospitals in the different regions have a detection routine [12]. Similar results are obtained in a clinical audit to establish the gap between practice and best practice in activities related to nutritional screening and assessment in New South Wales hospitals [13].

Although a wide range of tools, such as imaging, and functional and biological markers for malnutrition, are available, the objective measurement of the malnutrition domains is hampered by limitations intrinsic to the screening and assessment tools, such as interobserver variability, difficult reproducibility, technician experience, some tools are time consuming, other techniques are expensive, not all tools are validated, etc. Furthermore, the heterogeneity of the populations being evaluated, as well as the setting in which malnutrition is being investigated, impacts the definition of “gold standard” screening and assessment techniques being systematically adopted.

The aim of this review is to show the most widely used methods for nutritional screening to identify individuals at risk of malnutrition with different diseases, and the methods then used for the assessment of the nutritional status of the at-risk patients.

## 2. Methods: Literature Search Strategy

This is a literature review about nutritional screening and nutritional assessment tools. The bibliographic survey was carried out in the following databases: Publisher Medline (PubMed), Cochrane Library, Embase, and Web of Science (WOS). For the search, descriptors were identified in the Medical Subject Headings (Mesh), available from the US National Library of Medicine (http://www.nlm.nih.gov/mesh/, accessed on 1 April 2020). The descriptors used were “Nutrition Assessment”, “Nutritional status”, “Assessment of nutritional status”, “Nutrition screening”, and “Nutrition screening tools”, which were combined through the Boolean OR and AND operators. There was no restriction on the year of publication of the studies, so that there was no loss of important data.

The eligibility criteria were review, systematic review, meta-analysis, original studies, adults and/or elderly patients (aged over 18 years), and written in English or Spanish. A lateral search was also conducted, whereby the reference lists of relevant articles were searched for additional publications.

## 3. Early Diagnosis of Malnutrition: Nutritional Screening

Malnutrition continues to be an under-recognized, under-diagnosed, and, hence, under-treated problem. Therefore, it must be detected early and quickly, in order to put in place re-nutrition interventions and/or treat the underlying causes or contributory factors [14].

Nutritional screening is defined in a similar way according to both the American Society of Parenteral and Enteral Nutrition (ASPEN) [15,16] and the European Society for Clinical Nutrition and Metabolism (ESPEN) [1]: as a process to identify an individual who is malnourished, or at risk of malnutrition, to determine if a detailed nutritional assessment is required.

Nutritional risk detection tools are of major help in the daily routine to detect potential or manifested malnutrition in a timely fashion. These tools should be quick and easy to use, economical, standardized, and validated. Screening tools must be sensitive, specific, and reproducible. They should be applied in the first 24 to 48 h after admission and, in view of the nutritional deterioration associated with time in hospital, be repeated at regular intervals [17]. Screening methods must include at least three aspects: involuntary weight loss, inadequate nutrition, and the individual’s functional capacity. They should also include the existence of disease-associated metabolic stress.

The choice of screening method depends on the available infrastructure and resources, the possibility of automation, and the healthcare setting, among others. Thus, the European Society of Parenteral and Enteral Nutrition (ESPEN) generally recommends using Nutritional Risk Screening 2002 (NRS-2002) in hospitalized patients, the Malnutrition Universal Screening Tool (MUST) at the community level, and the first part of the Mini Nutritional Assessment (MNA-SF) in the elderly population [18].

It is important that each screening method is only used for the particular patient groups in which its validity and reliability are demonstrated. Although there is no “gold standard”, validity was established by comparing different methods, such as anthropometric measurements; other more comprehensive assessment tools, such as the MNA and the subjective global assessment (SGA) form; or objective assessment by experienced professionals. Reviews of the validity and reliability of screening tools [17,19] conclude that more than one method should be used to assess nutritional status, as none of the current tools are sufficiently reliable to determine patients’ nutritional status in the range of different situations potentially encountered [20]. Depending on the screening tools used, the proportion of patients nutritionally at risk varies [21,22,23,24].

## 4. Nutritional Screening Tools

### 4.1. Mini Nutritional Assessment Short Form (MNA-SF)

MNA-SF is the short form of the MNA used in nutritional screening. Full form (see below) is used for nutritional assessment. This short form includes only six elements that demonstrate the greatest consistency, sensitivity, and specificity in relation to the full form of the MNA and conventional nutritional assessment. Therefore, it is faster and easier to perform than the full version. It includes food intake issues, weight loss, mobility, the existence of acute disease, neuropsychological stress, and BMI. If the total score is 11 points or less, out of a total of 14 points, the patient is at risk of malnutrition or is malnourished, and the full nutritional assessment version should be administered. According to its authors, 80% of patients rated as being at nutritional risk with this tool are malnourished according to the full nutritional evaluation [25].

It is a useful screening tool for elders, is associated with poor clinical outcomes, and is able to predict functional decline [26,27,28,29]. MNA-SF appears to be the most appropriate nutrition screening tool for use in older adults [30]. Available online: https://www.mna-elderly.com/sites/default/files/2021-10/MNA-english.pdf (accessed on 1 April 2022).

### 4.2. Malnutrition Universal Screening Test (MUST)

This tool was developed by the British Association for Parenteral and Enteral Nutrition (BAPEN) [31].

It classifies patients into malnutrition risk levels based on BMI, the existence of a history of involuntary weight loss, and the likelihood of future weight loss secondary to acute illness, conditioning the absence of food intake for more than 5 days. Each item is valued from 0 to 2 points as follows: body mass index (BMI) > 20 kg/m^2^ = 0; 18.5–20 kg/m^2^ = 1; <18.5 kg/m^2^ = 2; weight loss <5% = 0; 5–10% = 1; >10% = 2; acute illness and its relation to food intake in the following five days, absence = 0; presence = 2. Low-risk patients are classified = 0 points; medium risk = 1 point; and high risk ≥ 2 points.

MUST is a popular screening tool for all types of hospitalized patients [32,33,34,35]; ESPEN recommends its use at community level [18], and its reliability is similar to that of the MNA in screening for nutritional risk in geriatric populations [36]. It can predict the length of hospital stay, the possibility of being discharged to other hospitals or long-stay centers, possibility of readmission, and it can monitor progress once the nutritional intervention has begun. It is shown to be fast and reproducible [37,38]. Available online at https://www.bapen.org.uk/images/pdfs/must/spanish/must-toolkit.pdf (accessed on 1 April 2022).

### 4.3. Simplified Nutritional Appetite Questionnaire (SNAQ)

This tool was developed in the Netherlands. It consists of three questions: if there has been weight loss (more than 6 kg in the last 6 months, or more than 3 kg in the last month), loss of appetite, and if the patient required nutritional supplementation in the last month. The responses to each question are reported on a scale ranging from “very bad” to “very good”, with a final score of 1 to 5. A score of 2 indicates moderate malnutrition, and 3 or more points denote severe malnutrition [39].

SNAQ is quick and easy to implement, and does not require specialized equipment. (Table 1).

### 4.4. Nutritional Risk Screening 2002 (NRS 2002)

NRS-2002 [40] was developed from 128 studies on the effectiveness of nutritional support geared towards identifying under-nourished patients who would probably respond adequately to nutritional support.

It has a preliminary phase with four questions: BMI < 20.5; weight loss in the last 3 months, reduced intake in the last week, and serious illness. If the respondent answers any of these questions in the affirmative, they go on to the screening phase. This phase takes into account, on the one hand, weight loss, BMI, and reduction in food intake, yielding a score of 0 to 3, and on the other hand, assesses disease severity, considering current clinical conditions, and chronic diseases with acute complications (major abdominal surgery, cerebrovascular accident, traumatic brain injury, or bone marrow transplant), also yielding a score of between 0 and 3 points.

The total score is obtained from the nutritional assessment and the severity of disease, and is age-adjusted in patients above 70 years. (+1 point). An NRS score < 3 indicates no risk of malnutrition, and an NRS score ≥ 3 indicates a high risk or clear malnutrition, and is an indication of the need for nutritional support. The NRS-2002 is evaluated and validated in several studies, including randomized controlled trials, and is shown to be reliable. It is the ESPEN-recommended screening tool for hospitalized patients [18]; it demonstrates high sensitivity and specificity when compared with the diagnosis of physicians experienced in malnutrition [41]; greater sensitivity and specificity is reported versus other screening tools in critically ill patients [42,43], and it shows an association with mortality, complications, and length of hospital stay in different studies [29,44,45]. (Table 2).

### 4.5. Malnutrition Screening Tool (MST)

Developed in 1999 by Ferguson et al., this is a quick and easy screening tool that includes questions about appetite, nutritional intake, and recent weight loss. A score of equal to or greater than 2, out of a total of 7, suggests the need for a nutritional assessment and/or intervention [46].

It is recommended for hospitalized, outpatient, and institutionalized adult patients [47]. (Table 3).

### 4.6. Nutrition Risk in the Critically Ill (NUTRIC Score)

This model was developed by Heyland et al., in 2011 to identify critically ill patients who are likely to benefit from an intensive nutritional intervention. The model seeks to integrate the absence of food intake, whether acute or chronic (recent reduction in food intake and hospital stay), inflammation (by means of interleukin-6, and the presence of comorbidities), nutritional status, and outcomes. It also includes the values of the Sequential Organ Failure Assessment (SOFA) and the Acute Physiology and Chronic Health Evaluation (APACHE II) [48]. It was subsequently modified (modified NUTRIC score), and the IL-6 value was removed, since the score presents similar validity and reliability without it [49] (see Table 4).

Patients with a high NUTRIC score who receive an adequate nutritional intervention have a lower incidence of complications than those in whom the nutritional intervention is not satisfactory, who have poorer survival outcomes. In a recent study with critical COVID-19 patients, this score successfully identified patients at high-nutritional risk [50]. ASPEN recommends the use of this score, as well as the NRS-2002, in critical patients, since its calculation takes both the patient’s nutritional status and disease severity into account [51]. The same conclusion on the validity of the use of the NRS-2002 and the NUTRIC score in critical patients is reached in a systematic review by Cattani [52], as well as by different studies in this type of patients [53,54,55].

### 4.7. Risk Scales Based on Nutritional Parameters

Screening tools include scales, which, rather than trying to classify nutritional risk, seek to ascertain the risk of the appearance of complications and patient mortality derived from nutritional parameters.

#### 4.7.1. Nutritional Risk Index (NRI)

The NRI is the oldest screening tool, and was initially described by Buzby et al., to examine the association between malnutrition and surgical outcomes [56].

It uses the following formula:Outcome = (0.363 × albumin) + (1.27 × (% weight loss)) + 0.119

A result of less than 2.71 is considered abnormal, and is associated with a complication rate of 27.5% and mortality of 22%, whereas patients with a higher value present rates of 14.6% and 2.8%, respectively.

A relationship is also found between this nutritional risk scale and hospital stay and, therefore, with hospital costs [57].

#### 4.7.2. Geriatric Nutritional Risk Index (GNRI)

This corresponds to a modification of the Nutritional Risk Index, adapted to geriatric patients [58]. It is regarded as an index of risk of morbidity and mortality associated with malnutrition, rather than as an index for the classification of malnutrition [59]. The prediction formula is:GNRI = (1.489 × albumin (g/L)) + (41.7 × (weight/ideal weight))

A score under 82 represents a high risk of complications, between 82 and 92 points to a moderate risk, and above 92, a low risk. In geriatric patients, this index is associated with complications and outcomes in different types of patients: postoperative patients, patients with heart failure, cancer, and chronic kidney disease, among others [60,61,62,63,64]. Together with the MNA, it is the most widely used index in elderly hospitalized patients [65], and is a useful clinical predictor of a poor six month outcome, although its accuracy of prediction is low [27].

#### 4.7.3. Prognostic Nutritional Index (PNI)

This was developed by Mullen et al., investigating the relationships between nutritional status and outcomes in surgical patients [66].

The formula is as follows:PNI% = 158 − (16.6 × albumin(g/L)) − (0.78 × (TSF)) − (0.20 × (TFN)) − (5.8 × (DH))
where TDF = triceps skinfold, TFN = serum transferrin, and DH = cutaneous delayed hypersensitivity to antigens.

Patients are classified as high-nutritional-risk with PNI >50%, as moderate between 40% and 49%, and as low-risk below 40%, with a significantly higher rate of complications and mortality in patients with high-nutritional-risk who do not receive a nutritional intervention in relation to those who do, or who have a low-nutritional-risk [67,68,69,70,71,72].

#### 4.7.4. Prognostic Inflammatory and Nutritional Index (PINI)

Initially applied to critical patients, in whom it proved to be a sensitive and specific marker of nutritional and inflammatory status, it was later applied to other types of patients, such as surgical and hemodialysis patients [73].

Calculated as (alpha1-acid glycoprotein (a1-AG) × C-reactive protein (CRP))/albumin × transthyretin. A PINI score = <1 is considered normal. A Score >30 = high life risk, 21–30 = high risk, 11–20 = medium risk, 1–10 = low risk, and <1 = minimal risk.

### 4.8. Other Nutritional Screening Tools 

See (Table 5).

## 5. Nutritional Assessment

The objective of nutritional assessment, according to ASPEN [15], is to document the basic nutritional parameters, identify risk factors and specific nutritional deficiencies, determine nutritional needs, and to identify the medical, psychosocial, and socioeconomic factors that may influence the prescription and administration of nutritional support. For ESPEN [1], the nutritional assessment provides the basis for the diagnosis of malnutrition according to a clinical, psychological, social, and nutritional history, and a clinical examination that includes information on weight, height, BMI, body composition, biochemical data, calorie, protein, fluid, and micronutrient needs. The Academy of Nutrition and Dietetics indicates that nutrition assessment is a “systematic approach for collecting, classifying, and synthesizing important and relevant data to describe nutritional status related nutritional problems, and their causes.” It is an ongoing, dynamic process that involves not only initial data collection, but also reassessment and analysis of client or community needs, and provides the foundation for nutrition diagnosis and nutritional recommendations, including enteral and parenteral nutrition [101].

It differs from nutritional screening in the amount of information obtained by different means to reach a diagnosis of malnutrition and its degree or severity, and it can also be used to assess changes in nutritional status, and the response to the nutritional intervention applied [102].

Over time, different nutritional assessment methods have been used, some complicated and expensive, used mainly in research, and others more affordable, which could be applied in routine clinical practice. The “gold standard” must be sensitive and specific, in order to make the nutritional diagnosis, but also to predict outcomes in relation to nutritional status and show changes in relation to the individual’s re-nutrition [102].

The different methods for carrying out the nutritional assessment are described below.

### 5.1. Clinical Assessment

The patient’s medical records are a useful source for detecting risk factors for malnutrition. Risk factors include diseases that affect ingestion, gastrointestinal motility, digestion and absorption, diseases that cause increased losses, or situations in which requirements are increased due to an increase in energy expenditure and/or protein catabolism. It is important that medical records include demographic and socioeconomic data that may influence a patient’s nutritional status, such as family structure, educational level, marginalization, beliefs, and lifestyle. Information about the patient’s physical activity, as well as the type of work they do, is also necessary [103].

The clinical examination should be aimed at highlighting data that indicate muscle atrophy, loss of subcutaneous fat, hydration status, and the presence of signs that can guide to specific deficits. Recently, the “nutrition-focused physical examination” (NFPE) has been championed, which consists of a full-body physical examination to identify alterations related to malnutrition such as muscle mass, subcutaneous fat, the hair, the skin, the eyes, the oral cavity, the nails, edemas, ascites, and the patient’s overall appearance. Muscle loss can be observed, with loss of muscle size and tone in different muscle groups. Subcutaneous fat can be assessed by palpation of the orbital area, triceps, and iliac crest. The presence of edema can be evaluated in the same way. Inspecting the patient can point towards the presence of overall alterations, and to vitamin deficiencies associated with malnutrition, which can be ascertained through the inspection of the hair, lips, gums, teeth, nails, and skin. The disadvantages of this examination are that it can be greatly affected in critically ill patients, acute illnesses, and processes with active inflammation. In the same way, obesity makes assessment difficult, particularly the assessment of muscle mass [104].

Dietary history, including the patient’s eating habits, could highlight the possibility of global or specific nutrient deficiencies. The evaluation of macronutrients (fats, carbohydrates, and proteins) is just as important as micronutrients (vitamins, trace elements). Assessment of dietary intake is challenging, and all current methodologies come with their individual strengths and weaknesses. Innovative technologies to improve dietary assessment methods are emerging and seem promising. Conventional methods include food records (prospective) or 24 h dietary recall/diet history/food frequency questionnaires (retrospective). The appropriate method to use depends primarily on the main objective of the study, the level of detail required, and the resources available [105,106,107].

### 5.2. Anthropometry

Anthropometry offers the most portable, commonly applicable, inexpensive, and noninvasive technique for assessing size, proportions, and composition of the human body.

#### 5.2.1. Weight and Derived Indices

Body weight is the most commonly used body parameter in practice. Short-term variations usually reflect variations in fluid balance, and long-term changes reveal changes in body mass, although they do not give us an idea of body composition. Other related parameters are used, such as the relationship with ideal weight, percentage weight loss in relation to usual weight, and body mass index (BMI).

Involuntary weight loss in the previous three months is of value. A loss of 5% is considered as moderate, and 10% as severe. This parameter is clearly associated with morbidity and mortality [108].

This is an essential parameter for screening, nutritional diagnosis, and for the requirements calculation [109].

#### 5.2.2. Body Mass Index (BMI)

BMI is a parameter that relates weight to height (BMI = Weight(kg)/Height^2^ (m^2^). It is used for diagnosis of malnutrition and obesity. It is easy to calculate, applicable to all adults, and is internationally recognized. There are clear inverse relationships between clinical risk and BMI. Values between 18.5 and 20 are a nutritional risk (22 for the elderly) and below 18.5 is malnutrition (20 for the elderly). It correlates well with mortality and complications, but is not a good early marker of malnutrition [108].

#### 5.2.3. Circumference Measures and Skinfolds

Midarm circumference (MAC) and triceps skinfold thickness (TSF) are also parameters used in assessment of nutrition. MAC is measured at the midpoint between the olecranon and the acromion. It relates quite well to the body’s protein component, results, and response to nutritional support. It measures all tissue (bone, muscle, and fat), but if it is combined with TSF, it yields the arm muscle area (AMA) according to the Heymsfield equations: man = (MAC − πTSF)^2^ − 10/4π; woman = (MAC − πTSF)^2^ − 6.5/4π [110].

TSF correlates well with fat mass (FM), so other skinfolds, such as the subscapular, bicipital, and abdominal skinfolds, are used to a lesser extent. In addition, the measurement of the folds presents important limitations in terms of reproducibility and variability, due to edema or other common problems in clinical practice. MAC, AMA, and TSF values must be related to the percentiles of the population for age and sex. Falling between the 5th and 15th percentile implies moderate malnutrition, and below the 5th percentile means severe malnutrition [109]. Calf circumference has also been used, with values of <31 cm indicating loss of muscle mass, and it can be a good predictor of hospital readmission [111].

A recent study confirms the existing correlation of many of the above anthropometric data with length of hospital stay and the probability of patients returning to their regular residence on discharge [112,113].

Any reader who wishes to explore the most widely used anthropometric data in nutritional assessment further is referred to a comprehensive review by Madden [114].

### 5.3. Body Composition Methods

Body composition describes body compartments, such as fat mass, fat-free mass, muscle mass, and bone mineral mass, depending on the body composition model used (Figure 1). This type of nutritional assessment is more objective and precise than methods based on anthropometry [115].

The objective of this section is to introduce the different body composition analysis techniques that can be used [116].

#### 5.3.1. Bioimpedance Analysis (BIA)

This is a simple, inexpensive, and non-invasive method for estimating body composition. It is based on the conduction of an alternating electrical current through the human body. The current runs easily through tissues that contain a great deal of water and electrolytes, such as blood and muscle, while fatty tissues and bones are more resistant. Therefore, the greater the fat-free mass, the greater the body’s ability to conduct the current. BIA provides good information about total body water, body cell mass, and fat mass when corrected for age, gender, and race, using validated equations. However, it is not recommended in patients with fluid overload. Body composition parameters, such as fat-free mass (FFM) and fat mass (FM), are evaluated using formulas that include endurance, reactance, weight, height, gender, and race, and vary depending on the population studied [117,118].

It takes resistance and reactance into account to calculate the phase angle (PhA), meaning that this is dependent, on the one hand, on the capacitance of the tissues associated with cellularity, cell size, and cell membrane integrity, and on the other hand of the behavior of resistance, which depends mainly on tissue hydration. PhA is the most widely used bioimpedance parameter for the diagnosis of malnutrition and clinical prognosis, associated with cell membrane integrity and hydration. A cut-off value of 5° is used for the phase angle in women and in men, because PhA values <5° are associated with frailty, malnutrition, and clinically adverse outcomes, such as disability and mortality [119,120]. Conventional BIA is inexpensive, easy to use, readily reproducible, and a precise method for body composition analysis when using specific equations developed and cross-validated in populations with similar biological and clinical characteristics to those of the target population [121].

#### 5.3.2. Dual-Energy X-ray Absorptiometry (DEXA)

This is currently considered an accurate model for measuring body composition. It is used mainly in research, due to its high cost and low availability, in addition to exposing the patient to a certain amount of radiation. DEXA relies on radiological density analysis, and is a useful method for measuring the amount of bone mineral and soft tissue (fat and fat-free mass). It can be used by means of a full-body study or by regional studies, which also indicate the distribution of subcutaneous or visceral fat [115,122]. Body thickness, hydration status, and diseases with water retention (e.g., heart, kidney, or liver failure) can affect DEXA results. DEXA may overestimate muscle mass in persons with extracellular fluid accumulation, due to its inability to differentiate between water and bone-free lean tissue. Further research is needed to assess lean mass with this method [123].

#### 5.3.3. Computed Tomography (TC)

This technique makes it possible to quantify fat mass and fat-free mass, provides information about the distribution of subcutaneous and visceral fat, and makes it possible to estimate skeletal muscle mass. This method is used mainly in research, due to its restricted availability, cost, the time involved, and exposure to ionizing radiation. CT can produce a local or global high-resolution three-dimensional image of the human body from different angles of vision. The known attenuations of X-rays in fat and muscle tissue (Hounsfield units) allow these tissues to be defined and quantified. Due to its high-resolution, CT allows muscle quantity to be measured accurately. CT also provides valuable information on muscle quality by evaluating muscle density, a parameter related to intra- and extramyocellular lipid deposition [124]. This technique has the problem of the ionizing radiation it produces, so it must be used with few slices, it cannot be used repeatedly, and its use is recommended for reasons other than nutritional study. Together with MRI, it is regarded as the gold standard for the analysis of body composition [110,115]. In recent studies, a CT scan proves that many screening tools do not appropriately classify cancer patients with cachexia or sarcopenia [125]. However, another study comparing sarcopenia measured by CT with the MUST tool finds a higher correlation of MUST with postoperative complications than measurement by CT [126].

#### 5.3.4. Magnetic Resonance Imaging (MRI)

Together with the technique described above, it quantifies fat and fat-free mass, as well as their distribution. It is based on the different magnetic properties of chemical elements such as hydrogen, which produces images of the body’s soft tissue, permitting the quantification of tissues, fat, and muscle. Its advantage over CT is the absence of ionizing radiation, although the time needed for the acquisition of high-quality scans and post-acquisition processing further impedes the large-scale implementation of MRI [110,124].

#### 5.3.5. Densitometry

This technique assumes that the body is composed of fat and non-fat compartments, if we know total body density; if we know the density of muscle and fat tissue, we can subtract these two components. Air displacement plethysmography or water displacement hydrodensitometry can be used to determine body density. If we know body volume, through air or water displacement, and body weight, we can ascertain its density (body weight/body volume). Since the density of fat differs from the density of fat-free mass, both can be determined using this two-compartment model [115].

#### 5.3.6. Other Techniques

Dilution methods: These methods seek to determine total body water by the dilution of non-radioactive isotopes. It is based on the Fick principle, whereby the volume of distribution of a substance is obtained by dividing the amount of this substance present in the body by its plasma concentration.

Total body potassium: since potassium is found primarily intracellularly, and the natural isotope is present in a constant fraction, measuring potassium allows us to calculate total body cell mass.

Neutron activation, by irradiating the body with neutrons, induces the emission of a characteristic gamma radiation spectrum, by which body composition can be viewed from a molecular point of view. It is an expensive method that permits the quantification of individual elements such as nitrogen, calcium, sodium, potassium, phosphorus, carbon, hydrogen, and oxygen. Although this technique is able to give a very accurate estimation of overall skeletal muscle mass, high costs, radiation exposure, and technical difficulty substantially limit the implementation of this technique.

#### 5.3.7. Muscle Ultrasonography

This method is used to measure the thickness of subcutaneous fat, as well as the area of certain muscles, particularly the anterior quadriceps rectus, which highlight muscle loss, in situations of malnutrition and catabolism, and its improvement in re-nutrition processes [127]. The procedure is quite simple, although interpretations can be subjective and difficult to perform. It has the advantage of being able to assess the muscle from a quantitative and qualitative point of view, and it is an innocuous technique, although the alterations in hydration and the greater or lesser pressure exerted by the interoperators render it necessary to provide adequate training to the technicians that perform it [123,128]. Being radiation-free, muscle ultrasonography may be used frequently. In addition, the equipment is portable, which allows muscle mass to be estimated at bedsides [129].

### 5.4. Functional Examination

Functional assessment is a key component in the assessment of nutritional status and in the follow-up of nutritional interventions, given that loss of function is the rule in malnutrition, and recovery is a sign of nutritional improvement. The first nutritional assessment tool to include functional assessment is the SGA [130]. Since then, different assessment scales for activities of daily living have been used, particularly in the elderly, which can be found in an excellent review by Russell [131]. Different methods of functional examination include:Functional measurement of muscle strength is important, since protein and energy deficiency decrease muscle strength and power, and general physical condition. Muscle function tests are very sensitive to nutritional deficiencies and, therefore, to nutritional interventions as well. The most widely used test is dynamometry, which measures voluntary muscle strength (hand grip strength) and correlates well with nutritional status and results, as well as with the response to nutrition and the rehabilitation process. It is easy to perform and provides quantitative data that can be used in the diagnosis of sarcopenia; one diagnostic criterion is a manual compression force of <27 kg in men and <16 kg in women [132]. There is an inverse relationship between the pressure produced and the number of postoperative complications, length of hospital stays, and hospital readmission rate [133]. It is one of the diagnostic criteria for malnutrition for ASPEN [133];Respiratory function: the measurement of peak flow and FEV1 reflects respiratory muscle strength, related to catabolism and protein loss;Immune function: measures the cellular response to intradermal antigens. Situations of severe malnutrition led to anergy: a lack of response to antigens.

### 5.5. Laboratory Parameters

In clinical practice, laboratory markers are data, which have the advantage of flagging a possible nutritional alteration earlier and more objectively, since they are not subject to the subjective assessment of many screening tools, although their greatest disadvantage is that some of them behave as negative acute-phase reactants [134]. Different laboratory parameters include:Serum albumin is the most extensively studied protein in relation to malnutrition, and it is shown to be a good predictor of surgical risk [135,136]. However, due to its long half-life of 18 days, it reflects the severity of the disease and not of malnutrition in acute situations, behaving as a negative acute-phase reactant which, in inflammatory situations, causes a reduction in its synthesis, an increase in transcapillary losses, and an increase in degradation and dilution due to hyperhydration. However, it is a good nutritional indicator in chronic malnutrition. Serum albumin is often included in certain nutritional screening tools, particularly nutritional risk scores [110,137,138,139];Shorter half-life proteins, such as transthyretin (2 days) and transferrin (7 days), are also subject to the same distribution and influences of dilution as albumin, but may be better and more sensitive reflections of nutritional status. Transthyretin, also called prealbumin, is a good marker of malnutrition when there are no signs of inflammation [140], and it is a good data item for following evolution after a nutritional intervention, even when inflammation is present [137]. Normal values are between 20 and 30 mg/dL, a moderate degree of malnutrition is between 10 and 20 mg/dl, and severe malnutrition corresponds to values below 10 mg/dL. In different studies it is correlated with visceral and muscle proteins compared with studies using BIA and DXA [141]. The C-reactive protein (CRP)/prealbumin ratio, known to be a prognostic indicator of complications, is proposed for assessing the effect of inflammation on prealbumin levels [142];Creatinine reflects kidney function, but also correlates with muscle mass. Creatine is metabolized to creatinine at a steady rate, and it is related to the muscle mass. Its excretion in 24 h is used to calculate the creatinine height index CHI% = (urine creatinine in 24 h × 100)/ideal creatin uria index obtained from standard tables. Values of >30% indicate severe muscle depletion, values between 15% and 30% are moderate, and below 15% is mild [110];Another parameter measured in urine is 3-Methylhistidine (3MH), which fundamentally depends on muscle degradation, pointing to a decrease in situations of muscle mass loss, and to an increase in situations of stress-associated protein catabolism [143];Nitrogen balance can be useful in critically ill patients in whom nitrogen intake is known, and nitrogen losses through urine can be measured either directly using the Kjeldahl method, or by extrapolating it from the urine’s urea content. Although it is not exact, it can provide guidance in ascertaining protein catabolism and as an indication for intake [143];Other parameters, such as cholesterol and total lymphocytes, are also correlated with the degree of malnutrition [134,137,144].

## 6. Methods of Nutritional Screening and Assessment

### 6.1. Subjective Global Assessment (SGA)

SGA was developed by Detsky et al., in 1987 [145]. It includes the patient’s history (weight loss, changes in food intake habits, gastrointestinal symptoms, and functional capacity), a brief physical examination (verification of decreased muscle mass, subcutaneous fat, or appearance of ankle edema, sacrum, and ascites) and the physician’s overall assessment of the patient’s condition. Each patient is classified as well-nourished (SGA-A), suspected or moderately malnourished (SGA-B), or severely malnourished (SGA-C). It is a method recommended by ASPEN, and is widely used in hospitalized patients, particularly in cancer patients [146].

It is useful for making a nutritional diagnosis, but it probably does not adequately monitor the nutritional evolution of the patient after a nutritional intervention [102]. However, in a major study in Canadian hospitals, SGA, together with HGS, proves to be the most robust predictor of longer hospital stays, and the likelihood of readmission [147]. A systematic review concludes that it is a valid tool for both medical and surgical patients [148]. Another review that compares different tools for nutritional diagnosis in critically ill patients concludes that the SGA is one of the best tools for diagnosing malnutrition in the intensive care unit (ICU), although the association between nutritional risk and mortality is less clear in critical patients [149]. It was validated in medical, surgical, critical patients, patients with chronic renal failure and cancer, as well as in geriatric patients [150,151].

There are adaptations of this method, such as the Patient-Generated Subjective Global Assessment (PG-SGA), carried out by Ottery in 1996 [152], which has two components: the first is called the PG-SGA short form, which serves as a nutritional screening, and the second is performed by a professional, scoring each of the items, classifying malnutrition in the same way as the SGA, and making a triage depending on the score, which indicates the type of nutritional intervention that is necessary. It is a method that includes screening, assessment, monitoring, and triaging for interventions [153]. It is currently the method of choice in cancer patients [154,155,156,157]. Available online: https://nutritioncareincanada.ca/sites/default/uploads/files/SGA%20Tool%20EN%20BKWT_2017.pdf (accessed on 1 April 2022).

### 6.2. Mini Nutritional Assessment (MNA)

MNA was jointly developed and validated by the Center for Internal Medicine and Clinical Gerontology ( Toulouse, France), the Clinical Nutrition Program at the University of New Mexico (New Mexico, USA), and the Nestlé Research Center (Lausanne, Switzerland). Its objective is the early detection of the risk of malnutrition in elderly patients, in order to carry out an early nutritional intervention without requiring a specialized nutritional team [158].

It is the most widely used screening tool in both institutionalized and hospitalized geriatric patients, combining screening and evaluation characteristics [159]. It includes 18 items in 4 sections: anthropometry (weight, height, BMI, weight loss, mid-arm and calf circumference); general evaluation (lifestyle, medication, mobility and presence of acute stress, dementia, or depression); dietary assessment (number of meals, type of food, amount of fluids ingested, and autonomy in eating); and subjective assessment (self-perception of health and nutritional status), all of them relevant to the nutritional status of the elderly. Both the MNA (complete form) used for nutritional status assessment [158], and an abbreviated MNA (MNA-SF) used as a screening tool [25] are available. If the total MNA-SF score is 11 points or less, the patient is at risk for malnutrition, and the full version of the nutritional assessment should be administered. In the latter, over 23.5 points is regarded as an absence of malnutrition, a score between 17 and 23.5 means there is a significant risk of malnutrition, and under 17 points shows clear malnutrition. In general, patients with a score below 17 usually have weight loss and low albumin levels, requiring a nutritional intervention and an assessment to identify the causes of the malnutrition. Between 17 and 23.5 points, patients may not present weight loss or low albumin levels, but they are very likely to present a decrease in calorie intake that can be easily reversed with a nutritional intervention [160].

MNA is reproducible, easy to perform, user-friendly, cheap, and presents high sensitivity and specificity [161]. It correlates well with nutritional status and objective nutritional values, and can predict hospital outcomes in different types of patients [162,163].

Available online: https://www.mna-elderly.com/sites/default/files/2021-10/MNA-english.pdf (accessed on 1 April 2022).

### 6.3. ESPEN Criteria

This describes the minimum consensus-based criteria for the diagnosis of malnutrition, which are applicable regardless of the clinical setting and the etiology of the malnutrition. It indicates two options for diagnosing malnutrition. The first option is by means of a BMI < 18.5 kg/m^2^, and the second an involuntary weight loss of >10%, or >5% in the last 3 months, and one of the following: BMI < 20 in adults or 22 in the elderly, or a low fat-free mass index (FFMI) of <15 and 17 kg/m^2^ in women and men, respectively [108]. It was validated in hospitalized and outpatient patients, and compared to the NRS-2002 and MUST [32,164], demonstrating a relationship with the prediction of mortality of hospitalized patients at 3 months and 1 year [165] (Table 6).

### 6.4. AND/ASPEN Tool (ASPEN)

This is a similar tool to the SGA. It includes six items: a reduction in intake, weight loss, loss of muscle mass, loss of subcutaneous fat, localized or generalized accumulation of liquids, and decreased muscle strength measured by dynamometry. If the patient has two or more of these items, they are malnourished. The degree of malnutrition, moderate or severe, is classified in three different contexts: malnutrition in the context of acute disease, in the context of chronic disease, or in the context of reduced intake without an accompanying inflammatory state [133] https://aspenjournals.onlinelibrary.wiley.com/doi/10.1177/0148607112440285#table1-0148607112440285 (accessed on 1 April 2022).

This tool also correlates well with negative clinical outcomes such as mortality, length of hospital stay, complications, and hospital readmission [14].

### 6.5. Global Leadership Initiative on Malnutrition (GLIM)

GLIM diagnostic criteria were developed by consensus over a three year period (2016–2018) by the leaders of the most important clinical nutrition societies (American Society for Parenteral and Enteral Nutrition [ASPEN], European Society for Clinical Nutrition and Metabolism [ESPEN], Latin American Federation of Nutritional Therapy, Clinical Nutrition and Metabolism [FELANPE], and The Parenteral and Enteral Nutrition Society of Asia [PENSA]) [166,167,168,169].

GLIM follows a two-step process. The first step involves the use of one of the validated screening tools to ascertain the existence of nutritional risk. The second step is assessment for diagnosis of malnutrition and its severity.

GLIM criteria are comprised of three phenotypic and two etiological criteria. In order to diagnose malnutrition, a combination of at least one phenotypic criterion (involuntary weight loss >5% in the last 6 months, low BMI, or reduced muscle mass) and one etiological (reduced food intake/assimilation and metabolic status caused by disease) must be present in the patient. Its severity is classified as moderate or severe malnutrition, depending on the degree of weight loss, BMI value, or the degree of reduction in muscle mass (see Table 7).

GLIM criteria identify approximately 40% of hospitalized adults as cases of malnutrition, with a satisfactory validity criterion, and sensitivity and specificity above 80%, in line with SGA [170,171,172]. Other authors do not find such a high sensitivity, but do find a strong association with mortality and admission to a critical care unit [173]. The agreement of GLIM with other diagnostic tools is related to the screening tool chosen to perform the first step of the process [174], finding excellent concordance with SGA in critically ill patients with COVID-19 [175].

### 6.6. Resume of Nutritional Assessments Tools 

See (Table 8).

## 7. Discussion

Malnutrition affects large numbers of patients, particularly the very frail, such as elderly patients [176], patients with a chronic inflammatory process such as cancer or other kidney, respiratory, or heart diseases [177], and those with an acute inflammatory process, such as critical or surgical patients [178,179,180,181]. Different publications highlight the unfavorable consequences of malnutrition, either due to lack of intake, inflammation, or both causes acting simultaneously [182,183].

From the pathophysiological standpoint, fasting causes a catabolic process in which the body preferentially consumes its stores of fat to produce energy. This is accompanied by a small degree of protein catabolism, which, over time, brings about an alteration in body composition, which ultimately leads to loss of function, loss of quality of life, the development of infectious complications, and, if these patients contend with disease, an increase in complications. The other mechanism that can lead to a similar situation, and which often accompanies fasting, is the catabolism caused by the stress and inflammation that accompany both acute and chronic disease. This process of defending the organism produces an accelerated protein catabolism that leads to the loss of the lean mass that is metabolically active, with the aforementioned functional alteration developing more or less rapidly [1,184,185].

Sarcopenia is recognized as a nutrition-related condition that may be related to the aging process (primary sarcopenia); however, it may also result from pathogenic mechanisms (secondary sarcopenia) that are disease-related, activity-related, or nutrition-related [1]. This all leads to poor outcomes in the health, quality of life, morbidity, and mortality of patients, accompanied by a significant increase in healthcare costs [11,186,187,188]. For this reason, early detection must be a systematic objective pursued as soon as the relationship is established between the social or the healthcare system and the individual [189,190], as adequate nutritional intervention is shown to reduce mortality and complications in hospitalized patients [191].

In this work, we sought to review and present the different tools available for the early detection of patients whose characteristics make them at greater risk of malnutrition, using nutritional screening tools. Once these patients are identified, we apply nutritional status assessment techniques to make a more accurate diagnosis of the malnutrition and its severity. Finally, we should introduce a nutritional intervention in line with the individual’s needs, with the aim of improving health outcomes, and thereby reducing complications, mortality, and healthcare costs.

There is not a gold standard for nutritional screening or for a complete nutritional assessment [20,192]. Screening tools are the first step in the nutritional care process. Some may help detect nutritional risk, others may predict clinical outcome, others do both in defined populations. There is currently no general screening tool that can predict the clinical outcome in every patient group in all care settings, due to the heterogeneity of the disease within patient groups and treatment settings [146,193]. In relation to screening, different tools emerged, and continue to do so, for the purpose of improving sensitivity and specificity to identify patients with a nutritional risk. The tools most commonly used in the hospital setting are MUST, SGA, and NRS-2002; in the outpatient setting, MUST; and in the setting of residential care, the MNA-SF [194]. The Academy of Nutrition and Dietetics indicates that MST is the tool that should be used in any patient, regardless of age, clinical history, or place where it is performed, based on Table 9 [195].

With reference to the groups of patients, in cancer patients, SGA and PG-SGA are the most widely used tools [157,196,197,198,199], although some authors also find MNA-SF [200], MST [201], MUST [35], SGA, or NRS [202] useful. In acute hospitalized patients, the most commonly used tools are the NRS-2002 and the MUST score [203,204]. Some authors find MUST to be more sensitive in hospitalized patients [34,205,206,207]. In a study with medical and surgical patients, MUST is associated with mortality and PG-SGA, and also with prolonged stays and readmissions [208]. In critical patients, the most-used scores are the NRS-2002 and the NUTRIC [52,53,54]. In patients with chronic kidney disease, different scores are used, such as MUST, MNA, MST, and SGA, but weight fluctuation due to fluid retention, which affects weight and BMI, means that more specific scores were investigated, such as the nutritional impact symptoms (NIS) [90], which was validated against SGA [209]. In elderly patients, DETERMINE, SNAQ, MUST, and GNRI are recommended, but MNA-SF and MNA are the most validated tools [210,211,212,213,214,215].

There are different tools for nutritional assessment whose objective is to diagnose malnutrition and its severity:Body composition measurement tools are used mostly in research, although some of them, such as anthropometry and BIA, can be used in the clinical setting, supported by CT, DXA, and MRI;Initiatives for performing nutritional assessment through tools such as SGA, MNA, ESPEN criteria, AND-ASPEN, and GLIM are recommended by the scientific societies, are intended to reach an easier and faster diagnosis, and can be applied to a greater typology of patients;The NFPE, together with anthropometric and biochemical values, and particularly with function measurements, such as quality of life and dietary intake surveys, together with muscle strength measurements. Although it is costly in time, it can give a nutritional diagnosis, determine the severity of the malnutrition, and help to highlight specific vitamin and micronutrient deficiencies.

Nutritional assessment initiatives are essential for optimal nutrition care. It is important to choose and validate the most accurate tools to monitor the nutritional status to improve the quality of life of patients. The following methods are suggested for the assessment of nutritional status: assessment tools initiatives (SGA, MNA, GLIM...), physical examination, biochemical and inflammation markers, dietary assessment, functional data, and body composition methods [216].

## 8. Conclusions

Malnutrition is common in hospitalized patients, yet often remains undetected by medical staff. Nutritional assessment is the ideal process to identify patients requiring nutritional support, however, it is time consuming to complete. Nutritional screening tools are useful for the rapid and early identification of malnutrition, but need to be paired with nutritional assessment for accurate malnutrition identification.

The objective of this review was to provide an overview of the different nutritional screening and assessment tools, with the aim of drawing attention to the importance of making an adequate diagnosis of nutritional status to implement appropriate nutritional interventions early, and to reduce the complications associated with malnutrition.

## Figures and Tables

**Figure 1 nutrients-14-02392-f001:**
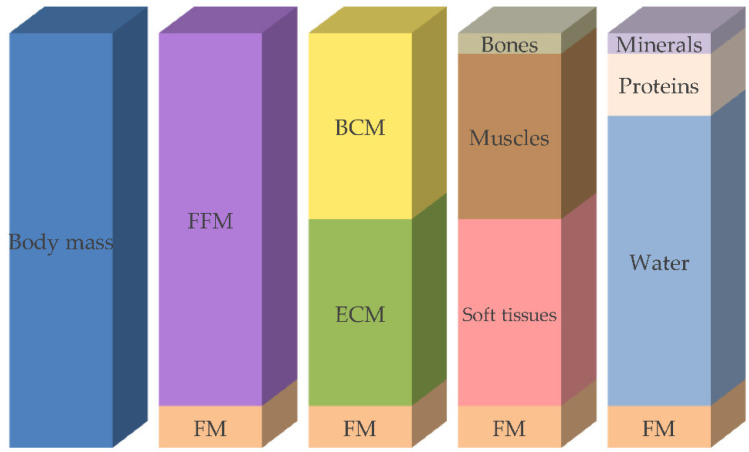
Compartment models of body composition. FFM: fat-free mass, FM: fat mass, BCM: body cell mass, ECM: extracellular cell mass. (Reber E, Gomes F, Vasiloglou MF, Schuetz Ph, Stanga Z. Nutritional Risk Screening and Assessment [Figure 1]. J Clin Med 2019; 8: 1065. Article licensed under Open Access Creative Commons Attribution License. https://www.mdpi.com/jcm/jcm-08-01065/article_deploy/html/images/jcm-08-01065-g001.png (accessed on 1 April 2022).

**Table 1 nutrients-14-02392-t001:** Simplified Nutritional Appetite Questionnaire.

Questions	Points
Did you lose weight unintentionally?	
More than 6 kg in the last 6 months	3
More than 6 kg in the last 3 months	2
Did you experience a decreased appetite over the last month?	1
Did you use supplemental drinks or tube feeding over the last month?	1

**Table 2 nutrients-14-02392-t002:** Nutritional Risk Screening (NRS-2002).

Impaired Nutritional Status		Severity of Disease (Stress Metabolism)	
Absent score 0	Normal nutritional status	Absent score 0	Normal nutritional requirements
Mild score 1	Weight loss 45% in 3 months or Food intake below 50–75% of normal requirement in preceding week	Mild score 1	Hip fracture; chronic patients, in particular with acute complications: cirrhosis; COPD; chronic hemodialysis, diabetes, oncology
Moderate score 2	Weight loss 45% in 2 months or BMI 18.5–20.5 + impaired general condition or Food intake 25–50% of normal requirement in preceding week	Moderate score 2	Major abdominal surgery; stroke; severe pneumonia, hematologic malignancy
Severe score 3	Weight loss >5% in 1 month >15% in 3 months or Body Mass Index of 18.5 + impaired general condition or Food intake 0–25% of normal requirement in preceding week	Severe score 3	Head injury; bone marrow transplantation; intensive care patients (APACHE 10)

Calculate the total score: 1. Find score (0–3) for impaired nutritional status (only one: choose the variable with highest score) and severity of disease (stress metabolism, i.e., increase in nutritional requirements); 2. Add the two scores (total score); 3. If age ≥ 70 years: add 1 to the total score to correct for frailty of elderly patients; 4. If age-corrected total =>3: start nutritional support.

**Table 3 nutrients-14-02392-t003:** Malnutrition Screening Tool (MST).

Have you lost weight recently without trying?	
No	0
Unsure	2
If yes, how much weight (kilograms) have you lost?	
1–5	1
6–10	2
11–15	3
>15	4
Unsure	2
Have you been eating poorly because of a decreased appetite?	
No	0
Yes	1
Total	
Score of 2 or more = patient at risk of malnutrition.	

**Table 4 nutrients-14-02392-t004:** NUTRIC Score.

Variable		Range	Points
Age		<50	0
		50–<75	1
		≥75	2
APACHE II		<15	0
		15–<20	1
		20–28	2
		≥28	3
SOFA		<6	0
		6–<10	1
		≥10	2
Number of co-morbidities		0–1	0
		≥2	1
Days from hospital to ICU admission		0–<1	0
		≥1	1
IL-6		0–<400	0
		≥400	1
Sum of points	Category	Explanation	
NUTRIC score scoring system, if IL-6 available			
6–10	High score	➢ Associated with worse clinical outcomes (mortality, ventilation). ➢ These patients are the most likely to benefit from aggressive nutrition therapy.	
0–5	Low score	➢ These patients have a low malnutrition risk.	
NUTRIC score scoring system, if no IL-6 available			
5–9	High score	➢ Associated with worse clinical outcomes (mortality, ventilation). ➢ These patients are the most likely to benefit from aggressive nutrition therapy.	
0–4	Low score	➢ These patients have a low malnutrition risk.	

**Table 5 nutrients-14-02392-t005:** Nutritional Screening Tools.

Tool/Acronym/Year	Features/Aspects	Patients Group	Reference
Instant nutritional assessment (INA, 1979)	Serum albumin levels and total lymphocyte counts	Cancer surgery, liver, and pancreatic diseases	Seltzer et al. [74]
Prognostic nutritional index (PNI, 1979)	Serum albumin, TSF, TFN, DH	Surgical patients	Mullen et al. [66]
Prognostic inflammatory and nutritional index (PINI, 1985)	C-reactive protein, orosomucoid, albumin, and transthyretin	Cancer patients, surgery, liver diseases, trauma, burn	Ingenbleek et al. [73]
Nutritional screening initiative checklist (DETERMINE, 1994)	Questionary about nutritional well being	Elderly people	Dwyer J. [75]
Nutritional Risk Index (NRI, 1988)	Serum albumin, current/usual body weight ratio.	All inpatients	Buzby et al. [56]
Malnutrition screening tool (MST, 1999)	Data about recent appetite status and weight loss	All inpatients	Ferguson et al. [46]
Risk Evaluation for Eating and Nutrition (SCREEN, 2000).	Factors affecting food intake, access to food, social factors, anthropometry, dietary intake	Elderly people	Keller et al. [76]
Malnutrition inflammatory score (MIS, 2001)	SGA method combined with BMI, serum albumin, and serum TIBC	Dialysis patients	Kalantar-Zadeh et al. [77]
South Manchester University Hospitals nutritional Assessment Score (2001)	Age, mental condition, weight, dietary intake, ability to eat, medical condition, and gut function	All inpatients	Burden ST [78]
Controlling nutritional status (CONUT, 2002)	Laboratory data (serum albumin, cholesterol, total lymphocytes, and hematocrit)	All inpatients	Ulibarri et al. [47]
Nutritional risk screening 2002 (NRS-2002, 2003)	BMI, weight loss, and acute disease score	All inpatients	Kondrup et al. [40]
Malnutrition Universal Screening Tool (MUST, 2004)	BMI, weight loss, and illness in relation to food intake	All inpatients	Elia et al. [31]
Rapid Screen (2004)	Weight change, BMI	Inpatients	Visvanathan et al. [79]
British nutrition screening tool (NST) 2004	Weight, height, recent unintentional weight loss, and appetite	All inpatients	Weekes et al. [80]
Simplified Nutritional Appetite Questionnaire (SNAQ, 2005)	Items related to appetite, food timing during day, food preferences, and daily number of meals	Elderly patients	Kruizenga et al. [39]
Geriatric Nutritional Risk Index (GNRI, 2005)	Serum albumin and the relationships between current weight and ideal weight	Elderly patients	Bouillane et al. [58]
Glasgow Prognostic Score (GPS, 2007)	Serum levels of albumin and C-reactive protein (CRP)	Cancer patients	McMillan et al. [81]
Protein Energy Wasting (PEW, 2008)	Serum chemistry, BMI, muscle mass, and dietary intake	Dialysis patients	Fouque et al. [82]
Cachexia consensus (2008)	Decreased muscle strength, fatigue, anorexia, low fat-free mass index, abnormal biochemistry	Cachexia diseases	Evans WJ et al. [83]
Mini Nutritional Assessment short form (MNA-SF, 2009)	First 6 items of 18 MNA	Elderly patients	Rubenstein et al. [25]
Imperial Nutritional Screening (INSYST, 2009)	Unintentional weight loss, reduced food intake	All inpatients	Tammam et al. [84]
3-Minute Nutrition Screening (3-MinNS, 2009)	Unintentional weight loss in the past six months, intake in the past week, body mass index (BMI), disease with nutrition risks, and presence of muscle wasting in the temporalis and clavicular areas	All inpatients	Lim et al. [85]
Objective screening nutrition dialysis (OSND, 2010)	Some anthropometric measurements, albumin, transferrin, and cholesterol levels	Dialysis patients	Beberashvili et al. [86]
Cancer cachexia classification (2011)	Weight loss, BMI, dietary intake, anorexia, muscle mass, metabolic change	Cancer patients	Fearon et al. [87]
Nutrition Risk in Critically ill (NUTRIC, 2011)	Age, APACHE II score, SOFA score, comorbidities, days in the hospital before admission to the ICU, and interleukin-6	Critically ill patients	Heyland et al. [48] Rahman et al. [49]
Spinal nutrition screening tool (SNST, 2012)	History of recent weight loss, BMI, age, level of SCI, presence of co-morbidity, skin condition, appetite, and ability to eat.	Spinal cord-injured patients	Wong et al. [88]
Royal Free Hospital Nutritional Prioritizing Tool (RFH-NPT, 2012)	Unintentional weight loss, BMI, influence of excess body fluids, and food intake.	Chronic liver disease	Arora et al. [89]
Nutrition impact symptoms score (NIS, 2013)	Symptoms impacting on food intake	Dialysis patients	Campbell et al. [90]
Eating Validation Scheme (EVS, 2013)	Eating habits	Elderly in primary care	Beck et al. [91]
Canadian Nutrition Screening Tool (CNST, 2015)	Weight loss, decreased food intake, body mass index (BMI)	All inpatients	Laporte et al. [92]
Royal Marsden Nutrition Screening Tool (RMNST, 2015)	Weight loss during the previous 3 months, a food intake of less than 50 % of normal in the previous 5 days, symptoms affecting intake	Cancer patients	Shaw er al. [93]
Malnutrition Inflammation Risk Tool (MIRT, 2016)	BMI, weight Loss, CRP	Inflammatory bowel diseases	Jansen et al. [94]
NUTRISCORE (2017)	MST, tumor location, active treatment	Cancer patients	Arribas et al. [95]
Saskatchewan Inflammatory Bowel Disease Nutrition Risk Tool (SaskIBD-NRT, 2018)	Weight loss, GI symptoms, anorexia, food intake restriction	Inflammatory bowel diseases	Haskey et al. [96]
BMI–lymphocyte–uric acid–triglyceride (BULT, 2019)	BMI, lymphocyte, uric acid, and triglyceride	Esophageal squamous cell carcinoma	Xu et al. [97]
Bach Mai Boston Tool (BBT, 2019)	Oral intake, body mass index (BMI), and weight loss in the last 3 months.	Cancer patients	Van et al. [98]
Dialysis Malnutrition Score (DMS, 2021)	Similar to PS-SGA with additional questions about dialysis history, and physical examination concerning loss of subcutaneous fat and muscle wasting.	Dialysis patients	Hassanin et al. [99]
Nutritional Screening inflammatory bowel diseases (NS-IBD, 2021)	BMI, unintended weight loss, GI symptoms, surgery for IBD	Inflammatory bowel diseases	Fiorindi et al. [100]

**Table 6 nutrients-14-02392-t006:** ESPEN Criteria.

Alternative 1:	BMI < 18.5 kg/m^2^
Alternative 2:	Weight loss (unintentional) > 10% indefinite of time, or >5% over the last 3 months combined with either: BMI < 20 kg/m^2^ if <70 years of age, or <22 kg/m^2^ if =>70 years of age, or FFMI < 15 kg/m^2^ in women and 17 kg/m^2^ in men

Two alternative ways to diagnose malnutrition. Before diagnosis of malnutrition is considered, it is mandatory to fulfil criteria for being “at risk” of malnutrition by any validated risk screening tool.

**Table 7 nutrients-14-02392-t007:** GLIM Criteria: Phenotypic and Etiologic Criteria for the Diagnosis of Malnutrition.

Phenotypic Criteria			Etiologic Criteria	
Weight Loss (%)	Low Body Mass Index (kg/m^2^)	Reduced Muscle Mass	Reduced Food Intake or Assimilation	Inflammation
>5% within past 6 months or >10% beyond 6 months	<20 if <70 years, or <22 if >70 years	Reduced by validated body composition measuring techniques	<50% of ER >1 week, or any reduction for >2 weeks or any chronic GI condition that adversely impacts food assimilation or absorption	Acute disease/injury or chronic disease-related

Available online at: https://www.espen.org/files/GLIM-2-page-Infographic.pdf (accessed on 1 April 2022).

**Table 8 nutrients-14-02392-t008:** Nutritional Assessment Tools.

Subjective Global Assessment (SGA, 1987)	Weight change, dietary intake change, gastrointestinal symptoms, functional capacity, and physical examination	Cancer patients, surgery, liver diseases	Detsky et al. [145]
Patient-Generated Subjective Global Assessment (PG-SGA, 1996)	Weight change, dietary intake change, gastrointestinal symptoms, functional capacity, and physical examination	Cancer patients, surgery, liver diseases	Ottery FD. [152]
Mini nutritional assessment (MNA, 1996)	Anthropometric measures, clinical history, and nutritional data	Elderly people	Guigoz et al. [158]
ASPEN Criteria for malnutrition (2012)	Insufficient energy intake, weight loss, loss of muscle mass, loss of subcutaneous fat, localized or generalized fluid accumulation, diminished functional status	All patients	White J et al. [133]
ESPEN criteria for malnutrition (2015)	BMI (<18.5 kg/m^2^), or weight loss and reduced BMI, or a low FFMI	All patients	Cederholm T et al. [108]
GLIM (2019)	Weight loss, BMI, muscle mass, dietary intake change, inflammation	All patients	Cederholm T et al. [166]

**Table 9 nutrients-14-02392-t009:** Validity of different screening tools.

Tool	Sensitivity	Specificity	Positive Predictive Value	Negative Predictive Value	Overall Validity	Agreement	Reliability
MST	Moderate	Moderate	Moderate	Moderate	Moderate	Moderate	Moderate
MUST	Moderate	Moderate	Moderate	High	High	Moderate	Moderate
MNA-SF	Moderate	Moderate	Low	Moderate	Moderate	Low	Moderate
SNAQ	Moderate	High	Low	High	Moderate	—	Moderate
MNA-SF-BMI	Moderate	Moderate	Moderate	High	Moderate	Moderate	—
NRS-2002	Moderate	High	Moderate	Moderate	Moderate	Moderate	—

## Data Availability

Not applicable.

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
