# Peer review of "Malnutrition Screening and Assessment"

_nutrients, 2022, doi:10.3390/nu14122392_

Round 1

Reviewer 1 Report

Line 54-59: ESPEN guidelines on definitions and terminology of clinical nutrition (Clin. Nutr. 2017) clarify this aspect subordinating to the general diagnosis of malnutrition the aetiology-based types of malnutrition. Guidelines describe definitions of disease-related malnutrition with or without inflammation, and malnutrition/undernutrition without disease. This concept should be integrated into the introduction. 

Line 84: what limitations are you referring to?

Line 167: on line 127-128 you have reported that MUST is recommended to use at the community level but in this section, you are referring only to the hospital setting. Must be clear 

Line 287: Definition of Nutrition Assessment should be integrated with the one reported by Academy of Nutrition and Dietetics (Nutrition Terminology Reference Manual (eNCPT): Dietetics Language for Nutrition Care). Nutrition Assessment is a systematic approach for collecting, classifying, and synthesizing important and relevant data to describe nutritional status, related nutritional problems, and their causes. It is an ongoing, dynamic process that involves initial data collection as well as continual reassessment and analysis of the client’s status compared to accepted standards, recommendations, and/or goals.

Line 315: consider a more recent bibliographic reference

Line 359: BMI cut-off is 18,5 

Line 382: translate

Line 506-507: the bibliographic reference is missing.  You could use the cut offs reported by the Sarcopenia: revised European consensus on definition and diagnosis (< 27 kg in men and < 16 kg in women)

Line 359: translate

Line 587: the official site is <pt.global.org>

Line 674: translate

Line 759: translate

Reviewer 2 Report

The authors reviewed the tools for screening malnutrition and assess nutritional status. The topic sounds interesting, but I have some concerns especially on the methods. Please fine the details below:

-Introduction: the description of the definitions of malnutrition is unclear, as well as the classification: I suggest to reorganize these paragraphs and to include both classifications acute/chronic and primary/secondary malnutrition and then specify on which types the article will focus on (making sure that the literature search is consistent); please avoid personal comments (“..we would agree”); I suggest to add % of incidence and/or prevalence for elderly and chronic diseases if available as the authors state they are the at-risk groups together with hospitalized (even if in real-life chronic disease and hospitalized patients may often overlap).

-Methods: I do not understand why a systematic approach according to PRISMA has not been implemented since the authors have done the effort to search into >2 databases. The number of retrieved articles should be added. If the authors choose to implement a systematic approach, then the abstract should be edited accordingly, and the PRISMA flowchart and checklist should be included.

-Results: I commend the authors for the thoroughness of the information provided. Anyway, I do think that the results should be reorganized since in my own opinion the screening tools and anthropometrics seem to be apart one from each other rather than both crucial part of the assessment. 

-Tables: some tables should be moved into supplementary material.

-I would include significant relevant references, such as: Nutrients 2020;  doi: 10.3390/nu12082413; Health Serv Res Manag Epidemiol 2021;. doi: 10.1177/23333928211064089. 

-Grammar mistakes and weird sentences are present (some Spanish word as well!). I recommend performing English editing of the text.

Reviewer 3 Report

Dear authors.

I found 31% repetition in your data. I am attaching the manuscript for your review and for amendments. 

Other than that, I feel that your manuscript did not add any novelty to us as researchers in the field of nutrition. It collected only the tools used to assess malnutrition. However, it facilitates the task searching for these kinds of tools. 

Author Response

Dear reviewer

I thank you your review of our work.
Taking into account that we have made a review, the repetition index should not be considered high. Definitions of some topics from different societies are cited verbatim as well as the content of the tables and, some statements correspond to conclusions of others.

I agree with you that we do not contribute any scientific novelty since our work is a narrative review. Our work is a compilation of tools aimed at fighting malnutrition and our goal is to draw attention to this problem. Probably our review is more educational and informative aimed at healthcare professionals who are not experts in nutrition.

I hope my answer satisfied you and you can forgive my not very good English.

Thank you very much. We are at your disposal.

Round 2

Reviewer 3 Report

Authors made changes and the manuscript